# Quantitative proteomic landscape of unstable atherosclerosis identifies molecular signatures and therapeutic targets for plaque stabilization

Yung-Chih Chen [1,2,3], Meaghan Smith[1], Ya-Lan Ying[1,2], Manousos Makridakis[4], Jonathan Noonan[1,2,3], Peter Kanellakis[1], Alin Rai [3,5,6], Agus Salim[7], Andrew Murphy[2,3,8], Alex Bobik[1,9], Antonia Vlahou [4], David W. Greening [2,3,5,6✉] & Karlheinz Peter[1,2,3,6✉]

Atherosclerotic plaque rupture leading to myocardial infarction is a major global health burden. Applying the tandem stenosis (TS) mouse model, which distinctively exhibits the characteristics of human plaque instability/rupture, we use quantitative proteomics to understand and directly compare unstable and stable atherosclerosis. Our data highlight the disparate natures and define unique protein signatures of unstable and stable atherosclerosis. Key proteins and pathway networks are identified such as the innate immune system, and neutrophil degranulation. The latter includes calprotectin S100A8/A9, which we validate in mouse and human unstable plaques, and we demonstrate the plaque-stabilizing effects of its inhibition. Overall, we provide critical insights into the unique proteomic landscape of unstable atherosclerosis (as distinct from stable atherosclerosis and vascular tissue). We further establish the TS model as a reliable preclinical tool for the discovery and testing of plaque-stabilizing drugs. Finally, we provide a knowledge resource defining unstable atherosclerosis that will facilitate the identification and validation of long-sought-after therapeutic targets and drugs for plaque stabilization.

[1] Atherothrombosis and Vascular Biology Program, Baker Heart and Diabetes Institute, Melbourne, VIC, Australia. [2] Central Clinical School, Monash University, Melbourne, VIC, Australia. [3] Department of Cardiometabolic Health, University of Melbourne, Melbourne, VIC, Australia. [4] Proteomics Research Unit, Biotechnology Division, Biomedical Research Foundation of the Academy of Athens, Athens, Greece. [5] Molecular Proteomics Laboratory, Baker Heart and Diabetes Institute, Melbourne, VIC, Australia. [6] Department of Cardiovascular Research, Translation and Implementation, La Trobe University, Melbourne, VIC, Australia. [7] Department of Bioinformatics, Baker Heart and Diabetes Institute, Melbourne, VIC, Australia. [8] Haematopoiesis and Leukocyte Biology Laboratory, Baker Heart and Diabetes Institute, Melbourne, VIC, Australia. [9] Department of Immunology, Monash University, Centre for Inflammatory Disease, School of Clinical Sciences, Monash Health, Melbourne, VIC, Australia. ✉email: David.Greening@baker.edu.au; Karlheinz.Peter@baker.edu.au

Cardiovascular diseases (CVD) are the leading cause of death worldwide[1]. The typical sequence of events starts with the abrupt rupture of unstable atherosclerotic plaques leading to occlusive thrombi and consequent myocardial infarction (MI). The molecular mechanisms contributing to the development of unstable plaques and, ultimately, plaque rupture remain largely unknown[2]. Difficulties in obtaining human unstable plaque tissue and, in particular, a lack of clinically relevant animal models, which adequately capture human plaque instability and rupture, so far limited the study of plaque instability.

Previously we developed a mouse model of hemodynamically driven atherosclerotic plaque instability (the Tandem Stenosis [TS] model; Fig. 1), which uniquely reflects the central features of plaque instability and rupture clinically observed in humans, such as thin fibrous caps, intraplaque hemorrhage, large necrotic cores, expansive vascular remodeling, neovascularization, and high inflammatory burden[3–9]. Utilizing this unique model, we now define the proteomic composition of unstable atherosclerosis in direct comparison to stable atherosclerosis and healthy artery tissue. We identify the networks that drive plaque instability (as distinct from stable atherosclerosis and vascular tissue), provide a comparative analysis of human plaque instability/rupture and identify potential therapeutic targets for plaque stabilization. The power of this approach is illustrated in the identification of fundamental neutrophil-derived molecular networks that drive plaque instability, including S100A8/A9. The role of S100A8/A9 in plaque instability was confirmed in human plaques. Its suitability as a therapeutic target for plaque stabilization was confirmed in the TS mouse model. Overall, our proteomic atlas of plaque instability provides unique insights into the pathology of unstable atherosclerotic plaques and represents a reliable preclinical platform for the discovery and testing of much-needed plaque-stabilizing drugs.

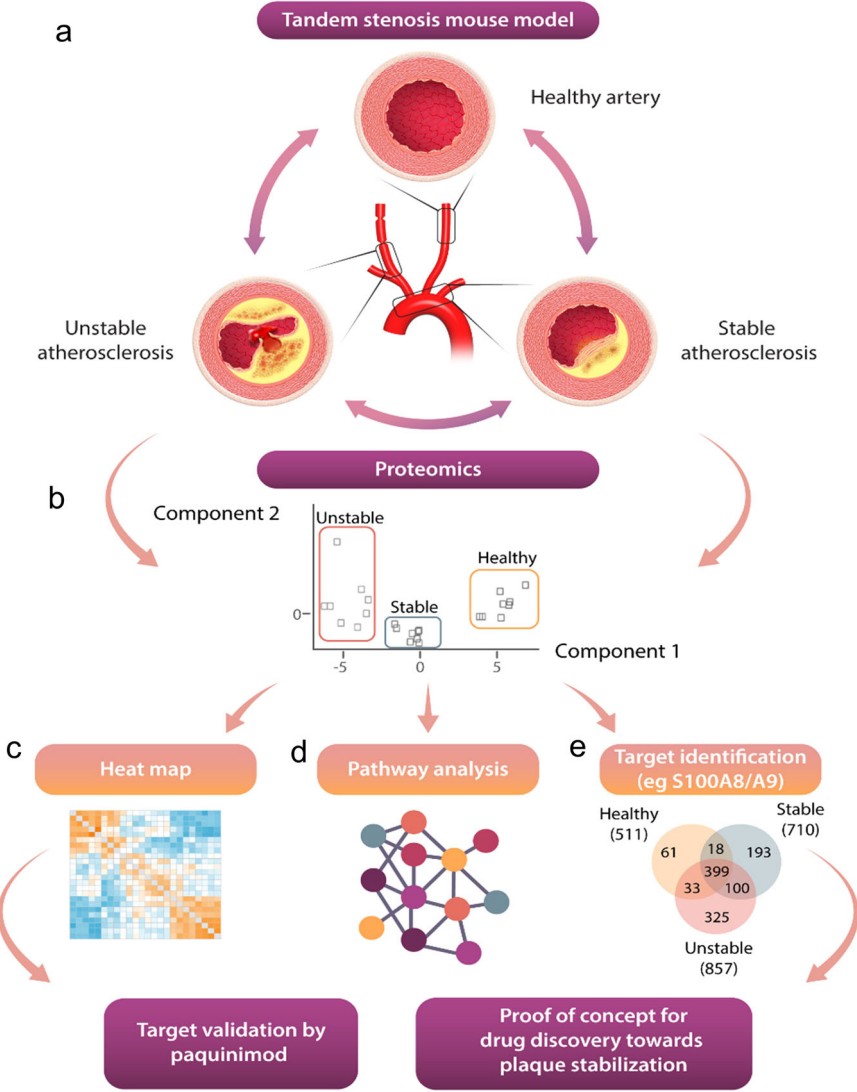

**Fig. 1 Workflow of quantitative proteomic comparison of unstable and stable atherosclerosis as well as healthy vessels using the tandem stenosis (TS) mouse model. a** Schematic workflow of proteomic analyses of vascular segments representing unstable, stable atherosclerosis (aortic arch, red), and healthy artery. Each segment (n = 15) was isolated and pooled for protein extraction, gel electrophoresis, and in-gel tryptic digestion. Single-shot label-free quantitative mass spectrometry was performed. The schematic drawing was created by somersault18:24 BV. **b** Principal component analysis and sample correlation matrix of distinct tissue groups (healthy arteries, stable plaques, unstable plaques). (**c/d/e**) Heat map, pathway/network enrichment analysis of unstable and stable plaque proteomes, and distributions of proteins identified across plaque compositions. Molecular target identification for plaque stabilization and validation in human plaques and the TS model.

## Results

### The proteomic landscape of unstable and stable atherosclerotic plaques.

We performed a comparative proteome analysis of TS-derived unstable atherosclerotic, stable atherosclerotic, and plaque-free healthy arteries using a label-free quantitative MS approach (Fig. 1a, Supplementary Fig. 1). Principal component analysis (PCA) demonstrated distinct proteomes in each tissue subsection (Fig. 1b). The left carotid artery was used as a healthy control as it does not develop atherosclerotic plaques despite systemic hypercholesterolemia as shown previously[8] and in Supplementary Fig. 2a, b. To reveal the similarity of biological replicates ($N = 8$ from 15 mice pooled per sample per phenotype), we used a sample-to-sample correlation matrix and confirmed correlations based on protein expression using hierarchical clustering (Fig. 1c). This finding points to an effective pooling strategy for acquiring sufficient tissue for proteome analysis. A total of 1130 quantifiable proteins were identified across unstable plaques (857 proteins), stable plaques (710), and healthy arteries (511). Moreover, for proteins uniquely identified in healthy arteries (61), stable plaques (193), and unstable plaques (325), these proteins were combined in downstream network/pathway analyses (Fig. 1d, e, Supplementary Data 1–2).

Hierarchical proteome clustering analysis of all artery phenotypes (proteins detected in at least 5/8 samples; one-way ANOVA, significance FDR < 0.05) shows 32 proteins abundantly expressed in diseased arteries (stable and unstable) compared to the healthy artery, while 53 proteins were minimally expressed in diseased arteries (Fig. 2a, Supplementary Data 3). Based on pathway enrichment analysis, compared to healthy arteries, diseased vessels (combined stable and unstable plaques) were enriched in several key processes and functions associated with "extracellular matrix (ECM) organization", "response to elevated platelet cytosolic calcium", "platelet activation, signaling and aggregation, integrin signaling", "fibrin clot formation", "platelet degranulation", and "RAF signaling" (Fig. 2b, Supplementary Data 3). Further unbiased analysis of the healthy vessel proteome (based on Supplementary Data 2) is provided in Supplementary Data 4, highlighting networks enriched ($p < 0.01$) in "metabolism," "muscle contraction," "ECM organization," "platelet activation, signaling, and aggregation," and "EPH–ephrin and ROBO receptor signaling." We show that, while all arteries are affected by systemic hypercholesterolemia, divergent pathways lead to plaque-free healthy arteries and diseased atherosclerotic arteries.

To understand specific pathological differences in the proteomic landscape between stable and unstable atherosclerosis, we combined uniquely identified and significantly differentially expressed proteins for further unsupervised hierarchical cluster analysis (unpaired Student's $t$-test, FDR < 0.05); this analysis included 253 stable plaque proteins (stable plaque proteomes) and 443 unstable plaque proteins (unstable plaque proteomes) (Fig. 3a & c, Supplementary Data 5). Specific pathways down-regulated in unstable plaques compared to stable plaques included "ECM organization," "TCA cycle and electron transport chain," "RHO GTPase activity," "cell–ECM interactions," and "smooth muscle contraction" (Fig. 3b, Supplementary Data 6). Further, pathway analysis using Reactome revealed upregulation of

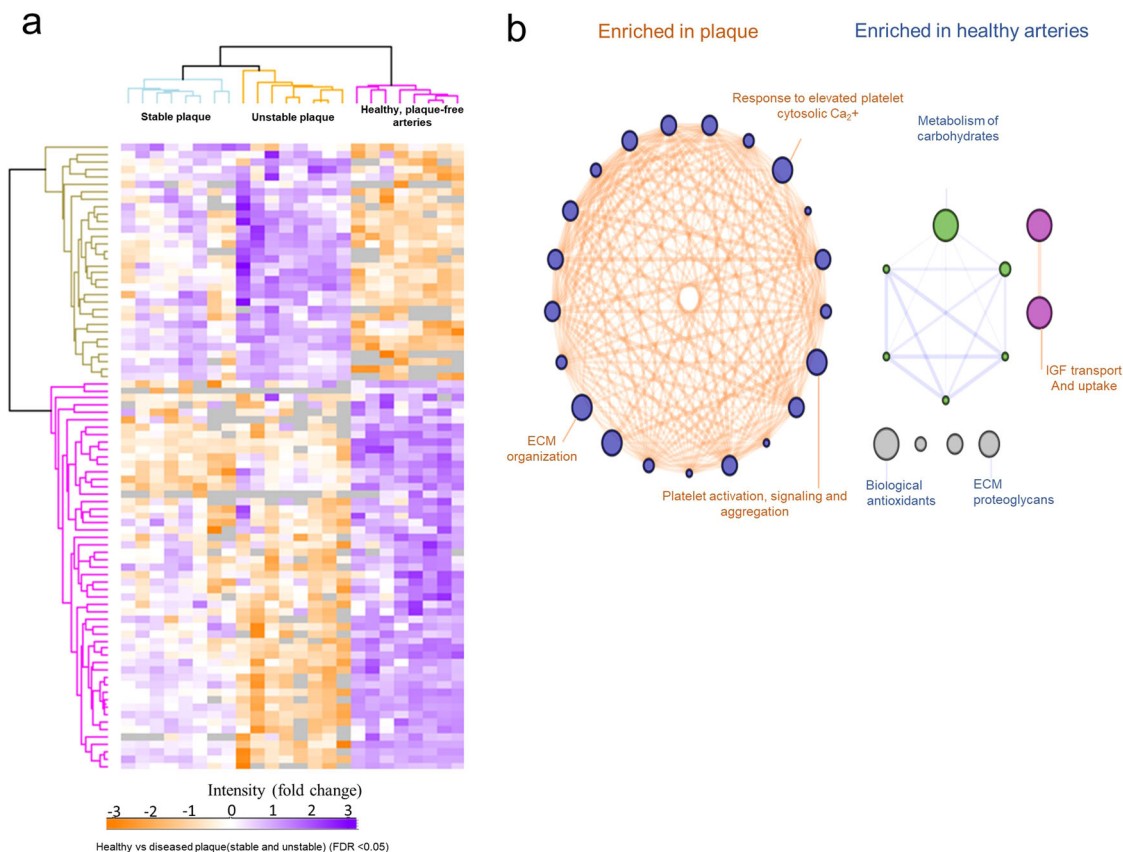

**Fig. 2 Differential proteome analyses of atherosclerotic (unstable and stable) arteries versus healthy, plaque-free arteries. a** Hierarchical clustering of differential protein expression across all tissue regions, comparing diseased plaque (stable and unstable) to the healthy vessels ($n = 8$). Proteins identified in at least 70% within each tissue group (at least 5/8 samples), FDR < 0.05, 32 proteins highly expressed in plaque (purple), 53 proteins lowly expressed in plaque (orange). **b** Enrichment map of Reactome pathway terms overrepresented in atherosclerotic plaques (stable and unstable) in comparison to healthy arteries.

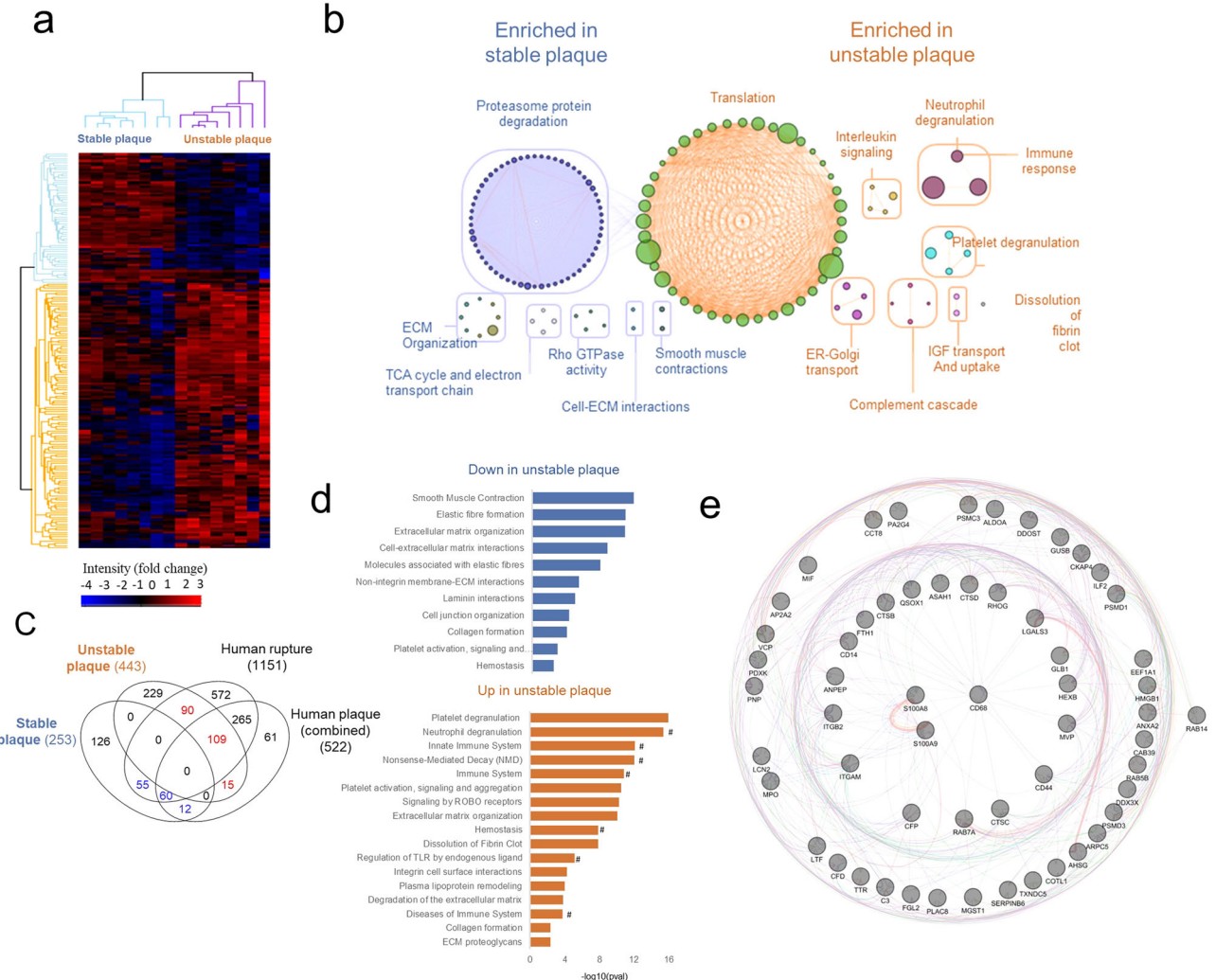

**Fig. 3 Defining the unstable and stable atherosclerotic plaque proteome. a** Global heatmap expression analysis between unstable and stable plaque (FDR < 0.05). **b** Enrichment analysis of Reactome pathways overrepresented in differential stable and unstable plaque proteome. **c** Comparative analysis of unstable and stable plaque proteomes with previous proteome analyses of human plaques by Vaisar et al., Hansmeier et al., Hao et al., and Liang et al.[10–13]. Proteins co-identified in stable plaques (blue) and unstable plaques (red) are highlighted. **d** Pathway involvement of indicated proteins is asterisk color-coded. Direct Reactome pathway analyses of proteins co-identified in stable plaques (blue) and unstable plaques (red) with human plaque data are shown, # denoting pathways associated with S100A8 and S100A9. **e** GeneMANIA-based radial interaction map of Reactome pathway enrichment analyses of stable and unstable plaque proteomes. Nodes represent proteins, and edges represent evidence-based direct physical interactions. S100A8/A9 is located in the center.

"translation," "immune response" (including neutrophil degranulation, platelet degranulation), "complement cascade," "interleukin signaling," "IGF transport and uptake," and "dissolution of fibrin clots" in unstable plaques (Supplementary Data 6). This comparative enrichment analysis reveals a distinct proteome landscape at the tissue level for unstable and stable plaques, including proteins associated with immune/inflammatory activity, neutrophils, and platelet degranulation in unstable plaques in contrast to a reduction in smooth muscle cell (SMC) contraction and ECM remodeling in stable plaque; such networks directly function in the regulation of structural integrity and plaque rupture.

**Differential TS plaque proteome reveals correlation with other mouse model's plaque proteomes**. To gain specific insights into the mechanism of plaque rupture, we compared our differential plaque tissue proteomes with existing mouse model plaque proteome studies (Supplementary Data 7)[10–13]. Firstly, we assessed the co-enrichment of proteins between our TS unstable plaques

and plaques from transgenic mice with macrophage-specific overexpression of urokinase (SR-uPA+/0), which also display several markers of plaque instability[13]. This investigation revealed that 10 proteins co-enriched in SR-uPA+/0 plaque were also identified in TS unstable plaque proteomes. These proteins function in apoptosis and senescence. (Supplementary Data 7). In addition, for proteins minimally expressed in murine aortic plaques (SR-uPA+/0) relative to healthy arteries (TS mice), we identified 11 proteins attributed to hemostasis function, and at a subcellular/cell-type level, associated with basal membranes and common fibroblast. Even though the genetic backgrounds of the two mouse models are different, the priming of plaque development was similar in both. The healthy arteries in our TS model displayed chronic endothelial activation and systemic hypercholesterolemia, but no plaque development.

**Human plaques correlate with TS unstable and stable plaque proteomes**. To gain a further understanding of the differential

stable and unstable plaque proteomes in the crucial context of translating our findings to human atherosclerotic disease and plaque instability/rupture, we correlated our data with reported human ruptured carotid endarterectomy proteome signatures[13]. Here, 252 proteins were uniquely identified in the stable plaque proteome, while 443 proteins were uniquely identified in the unstable plaque proteome (Fig. 3c, Supplementary Data 7). Using these uniquely expressed proteins, we highlight that 199 (90 + 109) out of 443 co-identified proteins (44%) of the unstable plaques correlate with human ruptures, whereas 124 (109 + 15) out of 443 co-identified proteins (27%) were correlated with human plaque combined proteome profiles. Further, 71 proteins out of 252 were co-identified proteins (28%) from stable plaques which correlated with human plaques combined, while 114 proteins out of 252 were co-identified proteins (45%) associated with human plaque ruptures (Fig. 3c, Supplementary Data 7)[10–13]. This study has discovered a substantial overlap in the proteomes of mouse and human plaques, emphasizing the relevance of using animal models to study human atherosclerosis. Because human carotid plaques are typically composed of a mix of unstable and stable plaques, Vaiser et al.[13] were able to dissect those with intraplaque hemorrhage as unstable plaques. Thus, we were particularly interested in the 109 proteins that were found in both the human plaque proteome signatures and the unstable (TS) plaque proteomes. These protein-generated pathways were associated with "platelet activation, signaling, and aggregation," "innate immune system," "immune system," "neutrophil degranulation," and "ECM organization" (Fig. 3d, Supplementary Data 7–8). In contrast, 59 proteins from stable plaques co-identified in both the human plaque proteomes and the pathways were associated with "smooth muscle contraction," "elastic fiber formation/function," "ECM organization/function/interaction," and "vasopressin" (including various collagens/myosins) (Supplementary Data 7-8). In this context, proteomics might be utilized to dissect/differentiate the unique contributions of both stable and unstable plaque proteome signatures, as well as to identify factors from human atherosclerotic plaques.

Supporting key changes at a molecular level of stable plaque included annotations associated with "smooth muscle contraction", "laminin interactions", "cell–ECM interactions", and "elastic fiber formation,"; highlighting the ECM for essential scaffolding and maintaining structural integrity contributing to plaque stability (Supplementary Data 8). Furthermore, the pathways associated with unstable mouse and human plaques include "platelet activation, signaling, and aggregation", "neutrophil degranulation", "immune system", "dissolution of fibrin clots", "ECM organization/degradation", "hemostasis," and "lipoprotein remodeling" (Fig. 3d, Supplementary Data 8). Despite different differentially expressed proteins for each tissue subset, several pathways were co-identified for unstable and stable plaque (and in human plaque, Fig. 3d). These include collagen formation, hemostasis, ECM organization, and platelet activation, signaling, and aggregation in both phenotypes of atherosclerotic mouse plaques (Supplementary Data 8). The importance of these pathways for atherosclerosis is obvious and individual proteins are up or downregulated, depending on their inhibiting or activating role in the particular pathway. For example, for collagen formation, we identified components specific to stable plaque (Col18a1, Col4a1, Col6a1, Col6a3, Ctss, Lox, Loxl1) in comparison to different components in unstable plaque (Col14a1, Col4a2, Ctsb, P4hb, Plec, Ppib, Serpinh1). These findings are consistent with the central but differential role of the various collagen types and differences in abundance in the pathogenesis of stable and particularly unstable atherosclerosis.

We have cross-validated the proteins differentially expressed in plaque stability/instability from mouse and human samples using this unbiased approach, supporting the concept that stable and unstable plaques have widely disparate proteomic signatures and that the underlying pathways of inflammation are the driving force of plaque instability/rupture across species.

**Pro-inflammatory response proteins S100A8/S100A9 in the unstable plaque proteome.** A salient finding between the unstable plaque with murine and human plaque composition was the unique identification of S100 family members S100A8 and S100A9 in unstable plaques of TS mice and human carotid plaques (based on fragmented spectra), and S100A8 in ruptured human plaques (Fig. 3e, Supplementary Fig. 3, Supplementary Data 7–8). During inflammation, S100A8/A9 is actively released and plays a critical role in modulating inflammatory responses[14]. We show S100A8/A9-associated functional enrichment categories included "neutrophil degranulation", "innate immune system", "immune system", "regulation of TLR by endogenous ligand", and "diseases of the immune system" (Fig. 3d, Supplementary Data 8).

**Immunostaining of S100A8/A9 in human and mouse plaque tissues.** Expression of S100A9 was validated using immunofluorescence staining and Western blot analysis of mouse plaques (Fig. 4a–c). Notably, S100A9 was only detectable in unstable plaques but not in stable plaques or healthy arteries, validating the tissue analysis at the proteome level (Fig. 4b, d, Supplementary Data 1, Supplementary Fig. 7). Further, using immunofluorescence S100A9 predominantly co-localized with myeloperoxidase supporting the finding that neutrophils are likely the primary source of these proteins as we have shown in other inflammatory disorders[15,16]. Moreover, we confirmed the expression of S100A8 and S100A9 in histological samples of human carotid atherosclerotic plaques ($n = 14$) (Fig. 4e–h).

**Inhibition of S100A9 stabilizes vulnerable plaques in the TS mouse model.** To further confirm the contribution of S100A9 to plaque instability/rupture, we applied a pharmacological approach using the S100A9 inhibitor ABR-25757 (paquinimod) (Supplementary Fig. 4)[15,17]. Based on the histological assessment, we show that ABR215757 did not influence lesion size (Supplementary Fig. 5a–c), collagen content (Supplementary Fig. 5d–f), necrotic core size (Supplementary Fig. 5g–i), or lipid accumulation (Supplementary Fig. 5j–l) in stable atherosclerotic plaques and therefore does not contribute to stable atherosclerosis.

Next, we examined the effect of S100A9 inhibition on plaque instability, assessing whether the local inflammatory status in unstable plaques in the TS mice was altered. We identified a significant decrease in CD68 staining, used to identify macrophages, in unstable plaques (Fig. 5a–c). Furthermore, the collagen content within unstable plaques was significantly higher in mice treated with ABR-215757 compared to the vehicle control (Fig. 5d–f). While the treatment with ABR-215757 resulted in a trend toward decreased necrotic core size in unstable plaques, the reduction was not statistically significant ($p = 0.09$) (Fig. 5g–i). Also, ABR-215757 did not change the plasma levels of total cholesterol, triglycerides, HDL and VLDL/LDL, or blood glucose levels in treated mice (Supplementary Fig. 6a–e). These findings indicate that ABR-215757, a pharmacological inhibitor of S100A9, has the potential to stabilize vulnerable atherosclerotic plaques by reducing macrophage infiltration and increasing collagen content.

## Discussion
The insidious nature of atherosclerosis renders it a major and typically unpredictable contributor to the high rates of CVD-

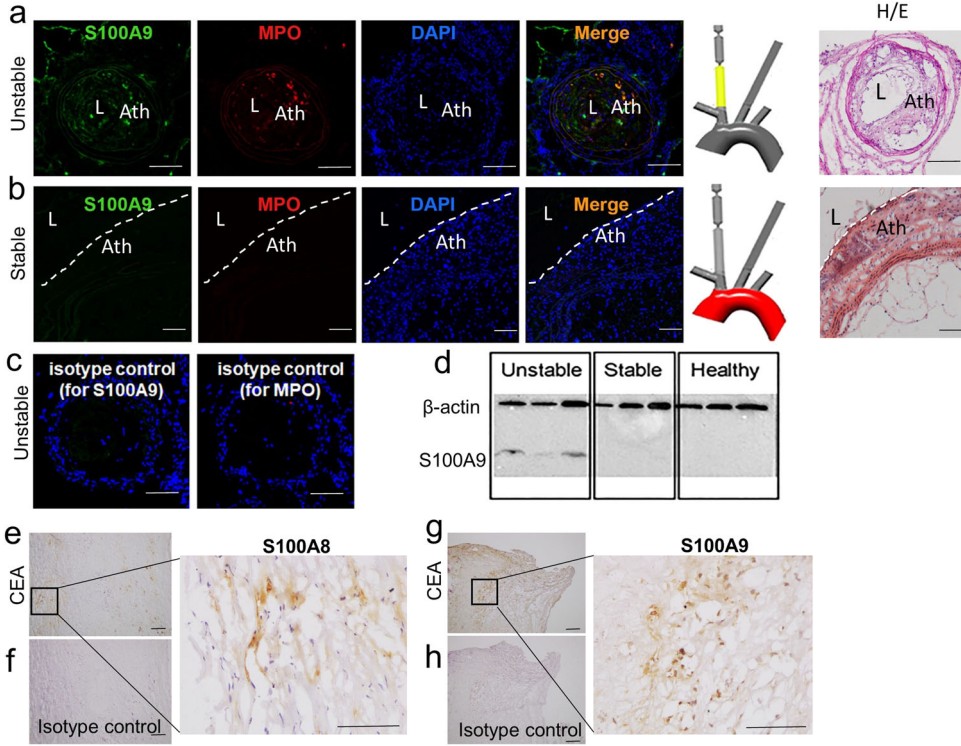

**Fig. 4 Validation of S100A8, S100A9 protein expression in TS plaques and human carotid plaques. a** S100A9, MPO, and DAPI immunofluorescence staining in unstable TS mice plaques ($n = 10$) and **b** stable plaques ($n = 10$). As a negative control, **c** IgG Isotype control antibodies for S100A9 and MPO were deployed. **d** Western blots were performed in three TS mice, and each artery segment was collected for protein identification. S100A9 is only found in the segments that are unstable. **e** S100A8 and **g** S100A9 immunohistochemistry in human carotid plaques ($n = 14$). As a negative control, IgG Isotype control antibodies for **f** S100A8 and **h** S100A9 were deployed. Bars indicate 100 μm. H/E: Hematoxylin and eosin, L: Lumen, Ath: Atherosclerosis. Dashed lines indicate endothelium.

associated mortality and morbidity globally[18]. Identification and, importantly, effective stabilization of vulnerable atherosclerotic plaques, which are at risk of rupturing and causing thrombotic events, are crucial issues in primary care and remain so for secondary prevention. In this study, we used a unique preclinical model of plaque instability/rupture, the TS mouse model[8], to identify the proteome signatures of unstable plaques in direct comparison to stable plaques and healthy artery tissues. Unlike traditional animal models of atherosclerosis, which typically develop stable atherosclerosis, the TS mouse model develops both stable and unstable, rupture-prone plaques at predefined areas in the vasculature. The unstable plaques in the TS model resemble the pathology seen in humans, including the development of thin fibrous cap ruptures, intraplaque hemorrhage, and intraluminal thrombosis[4–6]. These predefined areas of unstable and stable atherosclerosis in the TS model allowed the pooling of tissues originating from mice and building a comparable sample-to-sample correlation matrix and demonstrating the consistency of each individual pool sample cohort.

Importantly, unique proteins and pathways identified in the mouse plaque proteomes demonstrate significant similarity with human carotid plaque proteomes (a combination of four individual studies; 70% correlation across species)[10–13]. We focused on the 109 co-identified proteins that were more abundant in the TS mouse model of unstable plaques and in human plaque proteomes, including human rupture plaques[13]. These proteins constitute several protein families, including calprotectin, cathepsin, serine protease inhibitors, and components of the coagulation system. Calprotectin (S100A8/A9) is a known marker of neutrophil activation, degranulation, and NETosis[19]. The amount of serum S100A8/A9 correlates to the number of circulating

neutrophils, carotid artery disease, and other classic CV risk factors in middle-aged healthy individuals[15]. There are several proteases, including cathepsin B and cathepsin D, which are well-known potent collagenases and elastases that function through the degradation of the ECM and have been shown to cause plaque instability[20]. Interestingly, cathepsin D, in combination with S100A8/A9 and two other proteins, formed a 4-biomarker signature for CVD risk prediction over a 10-year follow-up[20]. In conjunction with the upregulation of serine protease inhibitors ITIH2 and ITIH4, our data indicate that unstable plaques contain an imbalance of proteolytic activity, potentially accelerating ECM degradation. Other imbalances, such as in the coagulation pathway, were also found in the unstable plaques. Proteins such as fibrinogen, prothrombin, and plasminogen were overexpressed in the unstable plaques, indicating the presence of (micro)thrombi at the time of sample collection. Microthrombi and intraplaque hemorrhage are well known to be strong makers of plaque instability in patients[21,22]. Our findings indicate that unstable plaques are dynamic in proteome composition in proteolytic, coagulative, and ECM degradation functions, the combination of these ultimately contributing to plaque rupture.

Using such co-identified proteins, our functional enrichment analyses revealed pathways downregulated in unstable plaques, including "smooth muscle contraction," and "ECM organization," The largest of these networks, "smooth muscle contraction," consists of the downregulated node proteins ACTA2. Not surprisingly, the protein ACTA2 is a marker of contractile SMCs in plaques. Shear stress may cause contractile SMCs to convert into synthetic SMCs, which can produce new collagen. A high ratio of ACTA2+ to CD68+ cells, along with thicker ECM, are more stable in late-stage atherosclerotic plaques[23]. Our findings are

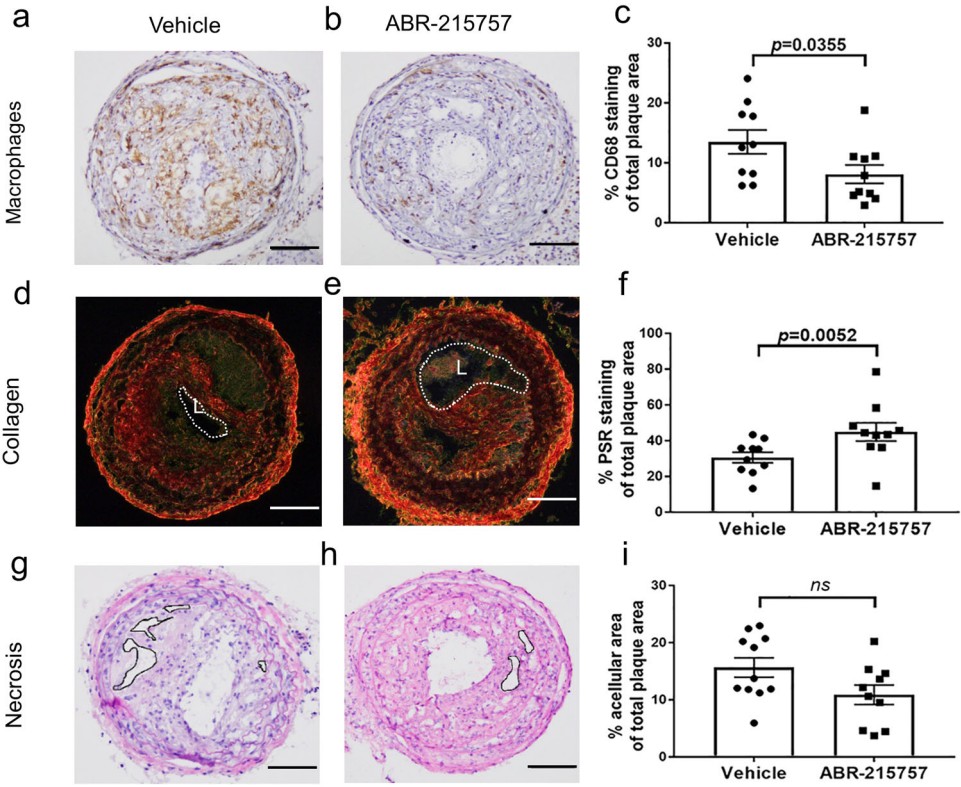

**Fig. 5 Inhibition of S100A9 stabilizes vulnerable atherosclerotic plaques in the TS mouse model.** ApoE$^{-/-}$ mice were given ABR-215757 (S100A9 inhibitor, 25 mg/kg, intraperitoneal injection) three times weekly for 7 weeks after TS surgery. **a, b** In mice treated with ABR-215757, TS mouse plaques showed a considerable reduction in CD68 foam cells. **c** CD68 quantification in percent of the atherosclerotic area. **d, e** Plaque collagen was detected with Picro Sirius red staining under polarized light, and **f** treatment with the inhibitor dramatically increased plaque collagen content. **g, h** H/E staining was utilized to define the necrotic core relative to the total plaque area. **i** Treatment with the S100A9 inhibitor showed a trend of lowering the necrotic core size. Data are presented as means and standard error of the mean (S.E.M.). The total plaque area refers to the plaque from the internal elastic lamina to the endothelium. Each dot represents the mean value of six sections from one mouse. L: Lumen. Student's t-test was used to test for statistical significance. Bars represent 50 μm.

consistent with the following statement published by the PlaqOmics Transatlantic Network: atherosclerotic plaques are destabilized by deleterious reprogramming of SMCs and other ACTA2 + fibrous cap cells, and these are critical biological differences determining susceptibility for coronary artery disease[23].

A potential limitation in our comparative proteomics approach is the fact that we cannot fully exclude the difference in baseline proteome between the carotid artery (source of the unstable plaque in the TS model) and the aortic plaque (source of the stable plaque in the TS model) confounds the comparison of the plaque stability state. However, the following three points will mitigate this risk. (1) In our analysis, we include a comparison with the plaque-free left carotid artery as a control. (2) We include a comparison to human plaques. (3) In the comparison between the carotid artery and aortic plaques, we do not see a major difference in the baseline proteomes, as the protein expression distribution and coefficient variations are not very different. We normalized our data at several levels to allow for comparison. Protein extracted from tissue was normalized (protein yield), and peptides used for sample analysis were further normalized following high-sensitivity peptide generation and cleanup, in addition to single batch post-processing informatic (TIC-based normalization using MaxQuant; maxLFQ). We provide protein expression distributions for all sample groups, where we did not observe baseline proteomic differences in individual samples between the aortic arch, carotid arteries with unstable plaque, and carotid arteries without plaque (Supplementary Fig. 1a). This is further supported by cross-group analysis of coefficient variation (Supplementary Fig. 1b), which did not differ

within our group, indicating that baseline proteomic variation (~20–22% coefficient variation) was similar across groups. In conclusion, the risk of the comparison of different localizations representing different phenotypes of plaques is a potential inherent confounding factor. However, we mitigated this risk as much as possible.

The major pathways upregulated in unstable plaques are associated with "neutrophil degranulation," "innate immune system," and "immune system". Using GeneMANIA functional interaction networks, we identified that S100A8 and S100A9, together with CD68, play a significant role in plaque inflammatory pathways. S100A8 and S100A9 are members of danger-associated molecular patterns which are known to initiate and promote inflammation and are predominantly expressed by neutrophils[24]. They form a heterodimer complex, which has been reported as a systemic biomarker for the detection of acute coronary syndrome[25,26]. However, contradictory results were reported in ApoE$^{-/-}$ and LDLR$^{-/-}$ mice[27,28]. In LDLR$^{-/-}$ mice, bone marrow S100A9 deficiency does not reduce atherosclerosis[27]. In ApoE$^{-/-}$ mice, whole-body S100A9 deficiency lowers atherosclerosis[28]. Nevertheless, neither animal model was used to evaluate the therapeutic potential of S100A9 inhibition in plaque stabilization. Importantly, in histological analyzes of unstable plaques in the TS model as well as in human carotid endarterectomy samples, S100A9 is highly expressed and co-localizes with neutrophils. Our findings are also in accordance with a previous report on a high abundance of neutrophils associated with rupture-prone plaques in humans[29].

We employed the small-molecule S100A9-inhibitor ABR-215757 in the TS model to validate our findings. Inhibition of S100A9 reduced CD68+ macrophage infiltration and increased collagen content in the fibrous cap of unstable plaques in the TS model. Our investigation is consistent with a prior report showing that ABR-215757 reduced diabetes-accelerated atherosclerosis in STZ-treated ApoE$^{-/-}$ mice, associated with reduced macrophage and lipid content[17]. Collectively, these promising effects on plaque stabilization support the development of ABR-215757 or similar drugs as therapeutics for the stabilization of atherosclerotic plaques, prevention of plaque rupture, and, ultimately, prevention of MI. The successful application of ABR-215757 as a plaque-stabilizing drug also supports the suitability of the TS model as a preclinical discovery tool for the identification of molecular targets and drug testing. The TS model has recently also been successfully applied to establish diagnostic technologies in relation to the identification of unstable plaques[5] and to demonstrate plaque stabilization via an MPO inhibitor[6] or SGLT2 inhibitor[30].

In conclusion, using quantitative proteomics in a preclinical TS mouse model, we established disparate natures and define the protein signatures of unstable and stable atherosclerosis. Integrating a robust proteome of unstable atherosclerosis derived from the TS model with existing murine data and in particular human plaque rupture, we define the core pathways contributing to unstable atherosclerosis, including neutrophil and platelet degranulation and inflammatory response pathways, while showing the core networks in stable plaques include SMC contraction and ECM remodeling. Confirming the TS model as a preclinical tool for drug discovery, with S100A8/A9, we identify a molecular target, validate this target in mice and humans, and demonstrate its suitability for therapeutic plaque stabilization. Finally, this study describes an atlas of numerous pathways and protein candidates to be tested for their suitability to identify/diagnose unstable plaques and, ultimately, develop long-sought-after plaque-stabilizing drugs.

## Methods

### Tandem stenosis (TS) surgery of the carotid artery and vessel segment dissection. Male ApoE$-/-$ mice from a C57BL/6 J background were obtained from the Animal Resource Center in Western Australia. The ApoE-/- were back-crossed to C57BL/6J for at least 10 generations. Mice at 6–7 weeks of age were fed on a high-fat diet (HFD) containing 22% fat and 0.15% cholesterol (SF00-219, Specialty Feeds, Western Australia) prior to TS surgery. After 6 weeks on an HFD, mice were anesthetized by intraperitoneal injection of 100 mg/kg ketamine and 20 mg/kg xylazine, and TS surgery was performed on all mice following procedures as previously described in detail.8 The stenosis was achieved by ligating the vessel with needles of specific diameters and then removing the needles.9 Post-surgical care and monitoring were provided for 48 h post-procedure, and mice were continued on an HFD for an additional 7 weeks before being culled by an overdose of ketamine and xylazine, and then perfused with 15 ml PBS buffer. The unstable plaque segments either with or without intraplaque hemorrhage, were identified under a dissecting microscope and recorded. All segments (unstable plaque: right carotid artery; healthy vessel: plaque-free left carotid artery; and stable plaque: aortic arch; Supplementary Fig. 2a, b) were isolated by dissection and snap-frozen in liquid nitrogen. The samples were stored at −80 °C until further use. All animal work was approved by the AMREP Animal Ethics Committee (E/1581/2015/B and E/1904/2019/B).

### Proteomics: tissue sample preparation. Detailed methods of protein extraction and proteomic analysis have been described previously[31] In brief, approximately 10 mg (net weight) of vessel samples from each of the pooled segment groups were homogenized in a lysis buffer (0.1 M Tris-HCl, pH 7.6) supplemented with 4% sodium dodecyl sulfate (SDS) and 0.1 M dithioerythritol (DTE), pulse centrifuged (16,000 g, 10 min), and the supernatant analyzed for protein concentration using a Bradford assay. Proteins (10 µg for each sample) were separated using SDS=polyacrylamide gel electrophoresis (PAGE, 4% stacking/12% separating gel), and the entire in-gel fraction isolated for analysis[32] Samples were reduced with 10 mM DTE in 100 mM NH4HCO3 at room temperature (RT) for 20 min, alkylated with 54 mM iodoacetamide for 20 min (in the dark) at RT, and digested with trypsin (600 ng/sample) at RT for 18 h (in the dark). Subsequently, peptides were extracted using 50 mM NH4HCO3 for 15 min at RT, followed by dilution with 10% formic acid (FA) and acetonitrile (I) (1:1), filtered (polyvinylidene fluoride, Merck Millipore), and lyophilized to dryness (SpeedVac centrifugal vacuum concentrator, Thermo Fisher Scientific). Peptides were acidified with a buffer containing 0.1% FA, pH < 3.

### Proteomics: liquid chromatography–tandem mass spectrometry. Peptides were analyzed using nanoscale liquid chromatography coupled to tandem mass spectrometry (nanoLC-MS/MS), where the peptides were loaded onto a nanoflow ultraperformance liquid chromatography (UPLC) instrument (Ultimate 3000 RSLS nano, Dionex) coupled online to an Orbitrap Velos FT mass spectrometer (Thermo Fisher Scientific) with a Proxeon nanoelectrospray ion source (Thermo Fisher Scientific)[33]. Peptides were loaded (0.1 × 20 mm 5 µm C18 beads, nano-trap column, Dionex; 5 µL/min in 0.1% FA, 2% ACN) and separated (Acclaim PepMap C18 nano-column 75 µm × 50 cm, 2 µm 100 Å, Dionex) at a flow rate of 300 nL/min at 35 °C. Liquid chromatography (LC) parameters: 480-min gradient from 1 to 65% (v/v) phase B (0.1% (v/v) FA in 80% (v/canACN); phase A (0.1% FA) (1–5% from 0 to 10 min, 10–25% from 10 to 360 min, and 25–65% from 360 to 480 min). The mass spectrometer was operated in MS/MS mode scanning from 350 to 2000 amu. The resolution of ions in MS1 was 60,000 and 15,000 for high-field collision-induced dissociation (HCD) MS2. The top 20 multiply charged ions were selected from each scan for MS/MS analysis using HCD at 35% collision energy. Automatic gain control (AGC) settings were 1,000,000 for full scans in Fourier transform mass spectrometry and 200,000 for MSn. Resolution in MS2 at m/z 115 was 16,300[34]. An MS1 scan was acquired from 350–2000 m/z (60,000 resolution, 1 × 106 AGC), 50 ms injection time) followed by MS/MS data-dependent acquisition of the top 20 most intense MS/MS ions from each scan, with HCD and detection in the orbitrap (resolution in MS2 at m/z 115 was 16,300, 35% normalized collision energy, 1.6 m/z quadrupole isolation width). Dynamic exclusion was enabled with a repeat count of 1, an exclusion duration of 30 s. Proteomic experiments were performed in biological replicates ($N = 8$ from 15 mice pooled per sample). Proteomic data (RAW and processed/search files) for each tissue region (healthy, stable, unstable) and comparisons between mouse tissue regions were uploaded to the Proteome Xchange Consortium via the PRIDE partner repository with the dataset identifier PXD030857.

### Proteomics: data processing and informatic analysis. Peptide identification and quantification were performed using MaxQuant (v1.6.6.0) with its built-in search engine Andromeda[35–37]. Tandem mass spectra were searched as a single batch against the Mus musculus reference proteome (UniProt; UP000000589, 59,345 entries, Feb-2019; canonical protein sequence) supplemented with common contaminants. Search parameters included carbamidomethylated cysteine as a fixed modification and oxidation of methionine and N-terminal protein acetylation as variable modifications. Enzyme specificity was set (C-terminal to arginine and lysine) using trypsin protease, with a maximum of two missed cleavages allowed. Peptides were identified with an initial precursor mass deviation of up to 7 ppm and a fragment mass deviation of 20 ppm. Protein identification was performed with at least one unique or razor peptide per protein group. Contaminants and reverse identifications were then excluded from further data analysis. "Match between run algorithm" in MaxQuant[38] and label-free protein quantification (maxLFQ) was performed, with proteins/ peptides matching the reversed database filtered out. The original MaxLFQ study by Cox et al.[39], is based on extracted ion current (XIC)-based approach and not spectral counting. This intensity-based approach takes into consideration the entire XIC for each sample for comparative purposes. The specifics of this algorithm are protected by the maxLFQ algorithms. However, the normalization approach has two main advantages (i) "delayed normalization," which makes label-free quantification fully compatible with any up-front separation, and (ii) extracts the maximum ratio information from peptide signals in arbitrary numbers of samples to achieve the highest possible accuracy of quantification. Perseus (v1.6.14)[40] was used to analyze proteins whose expression was identified across multiple biological replicates (i.e., in at least 70% in at least one group). Statistical analyses were performed using Perseus, R programming, and GraphPad Prism, with unpaired two-sample Student's t-test or one-way ANOVA performed (statistical significance defined at FDR < 0.05). Pathway enrichment map analysis was performed using Cytoscape (v3.7.1)[41], Reactome[42], and DAVID functional annotation[43] software; significance $p < 0.05$. Unique proteome profile compositions for each vessel segment were graphically visualized using Venny software[44]. Protein–protein interaction networks were described using StringApp incorporated into Cytoscape (v3.7.1)[45].

### Protein quantification and Western blotting. To obtain sufficient quantities of proteins from each segment, identical vessel segments from 4 mice were pooled together for each sample. To extract proteins from the vessels, the tissues were firstly homogenized in 1× RIPA lysis buffer (Cat# 20–188, Merck) with 1× cOmplete™ Protease Inhibitor Cocktail (Cat# 11697498001, Roche). Homogenized tissue suspension was then centrifuged for 20 min at 12,000 rpm at 4 °C. The supernatant containing the proteins was removed and placed in a fresh tube. A small amount of the sample was used to determine the protein concentration using a Pierce™ BCA Protein Assay Kit (Cat# 23225, Thermo Fisher Scientific) according

to the manufacturer's protocol. SDS-PAGE was performed to separate the proteins. Firstly, 15% separating gel with 4% stacking gel was prepared. Protein samples were denatured in Laemmli loading buffer (Cat #1610747, Bio-Rad) by boiling at 95 °C for 5 min. A total of 15 μg of proteins were loaded and separated on SDS-PAGE, then transferred onto a PVDF membrane. The membrane was blocked with 5% milk at RT for 1 h. Subsequently, the membrane was probed for anti-S100a9 primary antibody (1/5000, Cat # PA1-46489, Thermo Fisher Scientific) overnight at 4 °C, followed by incubation with HRP-conjugated anti-rabbit (1/5000, Cat# 205718, Abcam). The membrane was incubated with Pierce™ ECL Western Blotting Substrate (Cat# 32106, Thermo Fisher Scientific) for 5 min, and the protein bands were visualized using the ChemiDoc™ Gel Imaging System (Bio-Rad). After detection of S100a9, the membrane was stripped using Restore™ Western Blot Stripping Buffer (Cat# 21059, Thermo Fisher Scientific) and reprobed for β-actin (1/1000, Cat# 4970 S, Cell Signaling) as a loading control. Image J was used as the software to determine the densitometry of protein bands. The relative expression of protein candidates was normalized to the loading control β-actin.

**Immunofluorescence staining**. TS vessel segments were extracted and embedded in Tissue-Tek ® optimal cutting temperature compound (Sakura Finetek) and stored at −80 °C. Following the thawing of samples for 20 min at RT, samples were fixed in acetone at 20 °C for 10 min, followed by air drying. Samples were permeabilized in 0.1% Triton-X-100 in PBS for 10 min, followed by two PBS washes for 5 min at RT. Samples were blocked using normal serum blocker for 30 min at RT and then incubated in the primary antibody at 4 °C overnight. Samples were washed twice in PBS for 5 min and subsequently incubated in secondary antibodies Goat anti-Rat Alexa Fluor 488 (cat#A1106, Invitrogen) and Donkey Anti-Goat Tritc (cat#A16004, Invitrogen) for 30 min at RT. Samples were twice washed with PBS for 5 min, counterstained with Dapi (cat#D1306, Invitrogen), and mounted with Vectashield Mounting Medium (cat#H-1000, Vector). Negative isotype controls Rat IgG (cat#1-4000 Vector), and Goat IgG (cat#02-6202, Thermo Fisher) were performed in concurrence. Refer to antibodies in Supplementary Data 9.

**Pharmacological inhibition of S100A9 in the TS mouse model of plaque instability**. This study was designed to evaluate the effect of S100A9 inhibition using ABR-215757 on atherosclerosis plaque instability. ABR-215757 is a small-molecule inhibitor of S100A9. ABR215757 was dissolved in alkaline water at pH 9 and provided in solution at a stock concentration of 5 mg/ml. Vehicle control was prepared as alkaline water at pH 9 and autoclaved before administration in treatment. Male ApoE-/- mice at 6 to 7 weeks of age were fed an HFD containing 22% fat and 0.15% cholesterol (SF00-219, Specialty Feeds) for 6 weeks, as previously described. Following TS surgery, the mice were continued on an HFD and received either 25 mg/kg ABR-215757 (Active Biotech) or vehicle control via intraperitoneal injection 3 times weekly for a total period of 7 weeks. The mice were euthanized as previously described. At the termination of the study, body weights were recorded. Plasma was collected for lipid profile analysis as follows. Whole blood from the mice was collected using heparin as the anticoagulant and centrifuged at 300×$g$ for 10 min. Plasma was isolated from the mice for lipid profile analysis. Plasma was diluted with double distilled water in 1/5 dilution before analysis. Total plasma cholesterol and glucose concentrations were measured using an LX20PRO Analyzer (Beckman Coulter) in combination with the following kits: Cholesterol (Cat # 467825, Beckman Coulter), HDL Cholesterol (Cat # 650207, Beckman Coulter), Triglycerides GPO (Cat # 445850, Beckman Coulter), and Glucose (Cat # 442640 Beckman Coulter). The kits performed are based on a series of enzymatic colorimetric reactions. Colorimetric changes were measured at 520 nm for cholesterol and triglycerides, at 560 nm for HDL, and at 340 nm for glucose. Following perfusion, aortic sinuses and TS vessel segments of carotid arteries were collected for atherosclerosis histological assessment.

**Immunohistochemical quantification and assessment**. Mouse plaque tissue was embedded in Tissue-Tek O.C.T. compound (Sakura) and cryosectioned into 6 μm thick sections using a Cryostat Microm HM525 (Thermo Fisher Scientific). Sections were stained with primary antibody, rat anti-mouse CD68 (Cat#: MCA1957GA, Bio-Rad), followed by secondary antibody, mouse absorbed biotinylated rabbit anti-rat (Cat#: BA-4001, Vector Laboratories), followed by DAB substrate incubation (Supplementary Data 10) and counter-stained in hematoxylin before mounting in dibutyl-phthalate polystyrene xylene (DPX). Stained sections ($n = 6$ per mouse) were imaged using an Olympus BX61 microscope. The positively stained area of dark brown color was quantified using Fiji (Image J). Each image was the first color deconvoluted into H/DAB mode. Then the brown picture was selected, and the threshold was adjusted that matches the positive dark brown staining of the original image. The proportion of the total plaque area was indicated after drawing the region of interest from the internal elastic lamina to the vascular endothelium.

**Hematoxylin and eosin**. Frozen cryosections of aortic sinus and carotid segments I were thawed for 30 min prior to staining and subsequently rehydrated in distilled water for 5 min. The slides were incubated with Harris Haematoxylin (HH-500,

Amber Scientific) for 15 s and immediately washed in running tap water. Cryosections were incubated in alkaline water (sodium bicarbonate in tap water) for 10 s and then washed in distilled water, followed by staining with Eosin 1% alcoholic (EOS1-500, Amber Scientific) for 2 min. The slides were dehydrated with 95% ethanol for 3 min and then with 100% ethanol twice for 3 min each. Finally, the slides were cleared twice with xylene for 5 min each and mounted with DPX mounting medium (Thermo Fisher Scientific).

**Picro Sirius red (PSR) staining**. Frozen cryosections of aortic sinus and TS segments were thawed for 30 min prior to staining and then fixed in 10% neutral buffered formalin (Sigma-Aldrich) for 10 min. The slides were then washed twice in PBS for 5 min each, followed by staining with 0.1% picrosirius red staining solution conta12iriussirius red powder (Cat # 365548, Sigma Aldrich) in picric acid solution (Cat # FNNFF004, Fronine) for 1 h. The slides were then differentiated in 0.01 M hydrochloric acid, and adequate levels of differentiation were checked under a microscope. The slides were subsequently dehydrated with ethanol and cleared with xylene, as described above. Finally, the slides were mounted with DPX mounting medium (Thermo Fisher Scientific).

**Oil Red O staining**. Frozen cryosections were fixed in 10% neutral buffered formalin (Sigma-Aldrich) for 5 min and washed in PBS for 4 min, followed by washing in 60% isopropanol for 30 s. The slides were then stained with 0.6% Oil Red O staining solution containing Oil Red O powder (Cat # O0625, Sigma Aldrich) in 60% isopropanol for 1 h. Tissue sections were differentiated in 60% isopropanol and washed in distilled water for 2 min, followed by counterstaining with Harris Haematoxylin (HH-500, Amber Scientific). The slides were washed and finally mounted with Aquatex (Cat # 108562, Merck).

**Human carotid plaque candidate protein validation by immunohistochemistry**. Human carotid plaques were collected from patients who presented with symptoms such as stroke or transient ischemic attack and underwent endarterectomy in the operating theater in the Alfred Hospital, Melbourne, Australia (Ethics approval number: 130/110). Plaque tissue was embedded in the Tissue-Tek O.C.T. compound (Sakura) and cryosectioned using Cryostat Microm HM525 (Thermo Fisher Scientific). Cryosections were thawed at RT for 30 min and then fixed in acetone at −20 °C for 20 min. Subsequently, slides were treated with 3% hydrogen peroxide in methanol for 30 min to block endogenous peroxidase activity. Sections were incubated with 10% normal goat serum (Cat# S-1000, Vector Laboratories) for 30 min, followed by avidin and biotin blocking according to the manufacturer's requirement (Cat# SP-2001, Vector Laboratories). After the blocking steps, the sections were treated with primary antibodies (Supplementary Data 11) at 4 °C overnight. Sections were subsequently incubated with goat biotinylated anti-rabbit secondary antibodies (Cat# BA-1000, Vector Laboratories) for 30 min at RT, followed by the use of an avidin-biotin-peroxidase complex system (Cat#PK-4000, Vector Laboratories). Positive staining was developed using a 3,3′-diaminobenzidine (DAB) substrate kit (Cat#SK4100, Vector Laboratories) and visualized as a brown color on the sections. Then sections were counterstained with hematoxylin. Negative controls, including rabbit IgG isotype (Cat# I-1000-5, Vector Laboratories) control and primary antibody omission, were performed in parallel with the experiments.

**Statistics and reproducibility**. Label-free protein quantification intensities were log2 transformed. In vivo data were quantified using Fiji (Image J). Statistical analyses were applied using Student's $t$-tests or one-way ANOVA for parametric data and the Mann–Whitney $U$ test or Kruskal–Wallis test for non-parametric data. Normality testing was performed using the D'Agostino & Pearson normality test in GraphPad Prism. The experimental numbers ($n$) are listed in the figure legends. $n$ represents the number of biological replicates. Data are presented as mean ± SEM, with FDR and $p$-values less than 0.05 (as indicated) considered statistically significant.

**Reporting summary**. Further information on research design is available in the Nature Portfolio Reporting Summary linked to this article.

## Data availability

Data generated or analyzed during this study are included in this published article (and its supplementary information files) or available from data repositories. Proteomic data (RAW and processed/search files) for each tissue region (healthy, stable, unstable) and comparisons between mouse tissue regions are available from the Proteome Xchange Consortium via the PRIDE partner repository with the dataset identifier PXD030857. For proteomics analyses the Human Protein Atlas (https://www.proteinatlas.org/humanproteome/ tissue) and functional enrichment annotations using g:Profiler (https://biit.cs.ut.ee/gprofiler/) were used. Further pathway enrichment map analysis was performed using Cytoscape (v3.7.1)[41], Reactome[42], and DAVID functional annotation[43] software. Unique proteome profile compositions for each vessel segment were graphically visualized using Venny software[44]. Protein–protein interaction networks were described

using StringApp incorporated into Cytoscape (v3.7.1)[45]. Various human plaque datasets were compared[10–13]. Hierarchical clustering was performed in Perseus using Euclidian distance and average linkage clustering, with missing values imputed at z-score 0. R was also used for data analysis and data visualization (ggplot2, ggpubr packages). The uncropped gel for Fig. 4d is provided as Supplementary Fig. 7

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

## Acknowledgements

Yung-Chih Chen is supported by a Heart Foundation Future leader fellowship (No. 102068). Karlheinz Peter is supported by a National Health and Medical Research Council L3 investigator fellowship. David Greening is supported by Heart Foundation (Vanguard), NHMRC project grant (DG: #1139489, 1057741), Future Fund (DG: MRF1201805), Pankind Aust. Innovation Grant, and the Victorian Government's Operational Infrastructure Support Program.

## Author contributions

Y.C.C. contributed to the conception, acquisition of data, data analysis, and paper preparation. M.K.S. contributed to data analysis and paper preparation. Y.L.Y. contributed to acquisition of data, data analysis, and paper preparation. J.N. contributed to data analysis, paper preparation, and revision. P.K. contributed to the acquisition of data and data analysis. A.R. contributed to data analysis. M.M. contributed to acquisition of acquisition, data analysis and manuscript preparation. A.S. contributed to statistics, bioinformatics, and data analysis. A.M. contributed to the conception and paper preparation. A.B. contributed to the conception and paper revision. A.V contributed to conception, data analysis, and paper preparation. D.G contributed to data analysis and paper preparation, and revision. K.P. contributed to the conception, paper preparation, and revision.

## Competing interests

The authors declare no competing interests.
