## [Peer Review File · Communications Biology]

Reviewers' comments:

Reviewer #1 (Remarks to the Author):

Cardiovascular disease is the leading cause of death worldwide, and perpetuated by atherosclerotic plaques leading to occlusive thrombi and myocardial infarction. Atherosclerotic plaques may be characterized as stable or unstable depending on morphology and evidence of plaque rupture. The molecular mechanisms differing between stable and unstable plaques are largely unknown. The authors sought to use bioinformatics to decipher commonly expressed genes between the stable and unstable plaques compared to healthy tissue to identify the genes that are upregulated/downregulated and unique to different plaque states. S100A8/A9 was unique to unstable plaque identified in the initial filtering. S100A8/A9 is observed as a marker of inflammation and is associated with neutrophils. The S100A8/A9 inhibitor given to ApoE knockout mice on a high fat diet showed that it stabilized the plaques in the tandem stenosis in vivo model.

- Supplemental figure 2 is unreadable and valuable for the data to assess the mass spectrometry approach used prior to proteomic analysis.
- The tandem stenosis (TS) model displayed chronic endothelial activation and systemic hypercholesterolemia despite no plaque development in the healthy controls, would a high fat, high cholesterol diet rather than high fat only diet change the gene signature? High fat diet may contribute to differences observed from human samples/other mouse models. Supplemental cholate is generally used to increase triglyceride levels in mice, how would this change the gene signature?
- In figure 4, supportive data is warranted to show the plaque is present in the stable plaque images since minimal/no MPO is present and the plaque is not apparent in the image.
- ApoE knockout mice with whole-body deficiency of S100A8/A9 reduced atherosclerosis so is that why the inhibitor proved effective? How would this work in other models like LDLR knockout that didn't have reduced atherosclerosis?
- For many figures, images are blurry and text is hard to read
- Identical pathways are listed both in "up in unstable plaque" and "down in unstable plaque". What differences are observed within these pathways?
- Expansion of methods for quantifying/labelling in Figure 5 legend or supplemental methods would be helpful.

Reviewer #2 (Remarks to the Author):

The authors have leveraged their newly established and previously mouse model of unstable atherosclerotic plaque to conduct a proteomic analysis of unstable and stable plaques as well as normal vascular tissue. From their well powered mouse proteomic analysis, they identified multiple proteins that overlap with comparable conditions from human atherosclerotic plaques proteome analyses. Confirmation and biological validation were carried out on one particular target, S100A8/9, found to be significantly more prominent in the unstable plaque condition. Inhibition of this target promoted plaque stability in the carotid artery, confirming a possible mechanistic role of this protein and its associated inflammatory pathways in eventual plaque rupture. Prevention of plaque rupture could have wide reaching impact on major cardiac events, which remain the leading cause of death.

Overall, the experimental design is of high quality, the methods are very well articulated, and the interpretation / conclusions fit the data that have been presented.

The following are some minor comments:

- 1) Different arterial regions were used for each experimental group - and this may confound the comparisons in plaque state. What effect do baseline proteomic differences between aortic arch and carotid artery have on the comparison made between stable and unstable plaque (since stable plaques came from aortic arch while unstable are sampled from carotid artery)?

2) Please elaborate on details for the mouse reference proteome used, what database was this proteome extracted from (Uniprot?) and are the entries reviewed (by the magnitude perhaps not?) and do they include canonical sequences only or do they also include isoforms of proteins?

3)Supplementary methods - redundant entries for some of the methods.

4) Was normalization used in the maxLFQ algorithm, if so what method?

5) also in supplementary methods - line 183-185, perseus does not quantify the proteins, it is for post-processing, dataset filtering, statistics and visualization. Protein quantification is completed prior to data upload into perseus. Please clarify in the text.

6)Figure text for some panels is very small and hard to read - furthermore resolution of figures appears low and they are fuzzy.

Response to Reviewers' comments:

Reviewer #1

We thank the Reviewer for their encouraging comments. We have responded to the Reviewers' comments on a point by point basis. We believe that addressing the Reviewer's comments has further strengthened our manuscript.

Cardiovascular disease is the leading cause of death worldwide, and perpetuated by atherosclerotic plaques leading to occlusive thrombi and myocardial infarction. Atherosclerotic plaques may be characterized as stable or unstable depending on morphology and evidence of plaque rupture. The molecular mechanisms differing between stable and unstable plaques are largely unknown. The authors sought to use bioinformatics to decipher commonly expressed genes between the stable and unstable plaques compared to healthy tissue to identify the genes that are upregulated/downregulated and unique to different plaque states. S100A8/A9 was unique to unstable plaque identified in the initial filtering. S100A8/A9 is observed as a marker of inflammation and is associated with neutrophils. The S100A8/A9 inhibitor given to ApoE knockout mice on a high fat diet showed that it stabilized the plaques in the tandem stenosis in vivo model.

- Supplemental figure 2 is unreadable and valuable for the data to assess the mass spectrometry approach used prior to proteomic analysis.

Response: We acknowledge that the previously uploaded figures were not of sufficient resolution. We have now updated the resolution/layout of all figures following the Reviewers' and Editors comments.

- The tandem stenosis (TS) model displayed chronic endothelial activation and systemic hypercholesterolemia despite no plaque development in the healthy controls, would a high fat, high cholesterol diet rather than high fat only diet change the gene signature? High fat diet may contribute to differences observed from human samples/other mouse models. Supplemental cholate is generally used to increase triglyceride levels in mice, how would this change the gene signature?

Response: We thank the Reviewer for this comment. TS surgery was applied to the right carotid artery and the healthy control vessels used in our study is the non-surgical left carotid artery. We have previously demonstrated that the left carotid artery did not develop plaque despite systemic hypercholesterolemia (Figure 2, Chen et al. Circ Res 2013). Studies have shown that steady laminar shear is atheroprotective (Oren et al. ATVB 1998) and this could explain why the left carotid did not develop atherosclerotic plaques despite systemic hypercholesterolemia. In addition, the atherosclerotic plaque in ApoE^{-/-} mice preferentially developed in the inner curvature and bifurcation (Nakashima Y. et al., ATVB 1994) of the aorta, where turbulent flow was the strongest. The healthy carotid artery control used in the study was a segment of the vessel that experienced laminar shear and did not develop plaque. We used a high-fat, high-cholesterol diet (21% fat and 0.15% cholesterol without sodium cholate) to accelerate the development of lesions. This high-fat, high-cholesterol diet was designed to mimic the human Western diet and has been utilized by many laboratories in studies on atherosclerosis. We agree with the Reviewer that the ApoE gene deficiency, as well as the effects of a high-fat, high-cholesterol diet, may affect proteomic data. Therefore, to reduce these biases, such as global gene deletion and the dietary effects, our study compared plaque/tissue samples from the same mouse. This way we focus on the comparison between plaque-free tissue and stable and unstable plaques and thereby should negate the bias rightly mentioned by the Reviewer. To highlight this point we have now included the new figure in Supplementary Figure 2 A & B and the following text in the manuscript:

In line 78-80 page 4 “The left carotid artery was used as a healthy control as it does not develop atherosclerotic plaques despite systemic hypercholesterolemia as shown previously⁸ and in Supplementary Figure 2 A & B.”

Supplementary Figure 2: Gross anatomy 7 weeks after tandem stenosis surgery in ApoE^{-/-} mice. (A) The blue structure is a coated braided polyester suture. The white material represents atherosclerotic plaques. (B) A longitudinal section of the healthy carotid artery is also provided in H/E staining. The bar represents 50 μ m.

- In figure 4, supportive data is warranted to show the plaque is present in the stable plaque images since minimal/no MPO is present and the plaque is not apparent in the image.

Response: We appreciate the comments from the Reviewer. In this revision, we provide new images in Figures 4A and 4B. The new H/E images show carotid artery and aortic arch plaques, and we marked the lumen and atherosclerosis in the immunofluorescence images. We updated the figure legend to distinguish between the atherosclerotic area, the endothelial layer of the artery, and the location of the lumen.

Figure 4: Validation of S100A8, S100A9 protein expression in TS plaques and human carotid plaques. (A) S100A9, MPO, and DAPI immunofluorescence staining in unstable TS mouse plaques (n=10) and (B) stable plaques (n=10). As a negative control, IgG isotype control antibodies for S100A9 and MPO were deployed. (D) Western blots were performed in three TS mice and each artery segment was collected for protein identification. S100A9 is only found in segments that are unstable. (E) S100A8 and (G) S100A9 immunohistochemistry in human carotid plaques (n=14). As a negative control, IgG Isotype control antibodies for (F) S100A8 and (H) S100A9 were deployed. Bar indicates 100 μ m. H/E: Hematoxylin and eosin, L: Lumen, Ath: Atherosclerosis. Dashed lines indicate endothelium.

- ApoE knockout mice with whole-body deficiency of S100A8/A9 reduced atherosclerosis so is that why the inhibitor proved effective? How would this work in other models like LDLR knockout that didn't have reduced atherosclerosis?

Response: In our study, we also examined the function of an S100A9 inhibitor on stable atherosclerotic plaques occurring in the aortic sinus. We found that lesion size, collagen content, necrosis, and lipid deposition within the aortic sinus were not affected by the

S100A9 inhibitor (Supplementary Figure 4). Thus, the S100A9 inhibitor seems to only affect unstable plaques, but not stable plaques.

We have performed the TS model in LDLR^{-/-} mice and were able to induce unstable atherosclerotic plaques identical to ApoE^{-/-} mice. So far, we have not tested S100A9 inhibitors in LDLR^{-/-} mice. However, based on our experience in comparing ApoE^{-/-} mice with LDLR^{-/-} mice, we would not expect different outcomes with the TS model.

- For many figures, images are blurry and text is hard to read

Response: We apologize for the low resolution of the previously uploaded main and supplemental Figures. We have now updated the resolution/layout of all figures.

- Identical pathways are listed both in “up in unstable plaque” and “down in unstable plaque”. What differences are observed within these pathways?

Response: We thank the Reviewer for pointing this out. In both analyses (up and down regulation) these pathways were identified as exhibiting major differences. Differentially expressed proteins for unstable versus stable mouse plaque and also co-identified with stable human plaques were used for pathway enrichment analysis. Although these pathways were identified to be differentially regulated, both up and down regulated, the individual components being up or down regulated were distinct. We have now highlighted and color-coded identical pathways for each subset, as well as components of these differentially regulated pathways in Supplementary Table 8. We also followed the Reviewer’s comment and have provided further texts in the manuscript as below:

*Page 8: ...”Despite different differentially expressed proteins for each tissue subset, several pathways were co-identified in exhibiting major differences both in unstable and stable plaque (and in human plaque, Figure 3D). These include collagen formation, hemostasis, ECM organization, and platelet activation, signaling and aggregation in both phenotypes of atherosclerotic mouse plaques (**Supplementary Table 8**). The importance of these pathways for atherosclerosis is obvious and individual proteins are up or down regulated depending on their inhibiting or activating role in the particular pathway. For example for collagen*

formation, we identified components specific to stable plaque (Col18a1, Col4a1, Col6a1, Col6a3, Ctss, Lox, Loxl1) in comparison to different components in unstable plaque (Col14a1, Col4a2, Ctsb, P4hb, Plec, Ppib, Serpinh1). These findings are consistent with the central but differential role of the various collagen types and differences in abundance in the pathogenesis of stable and particularly unstable atherosclerosis.”

- Expansion of methods for quantifying/labelling in Figure 5 legend or supplemental methods would be helpful.

Response: In direct response to the Reviewer’s comment, we now provide the detailed methods for quantification in the legend of Figure 5. We also provide the detailed method for quantification in the Supplementary Material subheading “Immunohistochemical quantification and assessment” on page 8 to 9.

Figure legend for Figure 5:

Figure 5: Inhibition of S100A9 stabilizes vulnerable atherosclerotic plaques in the TS mouse model. ApoE^{-/-} mice were given ABR-215757 (S100A9 inhibitor, 25mg/kg, intraperitoneal injection) three times weekly for 7 weeks after TS surgery. (A, B) In mice treated with ABR-215757, TS mouse plaques showed a considerable reduction in CD68 foam cells. (C) CD68 quantification in percent of atherosclerotic area. (D, E) Plaque collagen was detected with picro sirius red staining under polarized light, and (F) treatment with the inhibitor dramatically increased plaque collagen content. (G, H) H/E staining was utilized to define the necrotic core relative to the total plaque area. (I) Treatment with the S100A9 inhibitor showed a trend of lowering the necrotic core size. Data are presented as means and standard error of the mean (S.E.M.). Total plaque area refers to plaque from the internal elastic lamina to the endothelium. Each dot represents the mean value of six sections from one mouse. L: Lumen. Student’s t-test was used to test for statistical significance. Bars represent 50 μ m.

Page 8 to 9(Supplementary material):

“Immunohistochemical quantification and assessment

Mouse plaque tissue was embedded in Tissue-Tek O.C.T. compound (Sakura) and cryosectioned into 6 μ m thick sections using a Cryostat Microm HM525 (Thermo Fisher Scientific). Sections were stained with primary antibody, rat anti-mouse CD68 (Cat#: MCA1957GA, Bio Rad)

followed by secondary antibody, mouse absorbed biotinylated rabbit anti-rat (Cat#: BA-4001, Vector Laboratories), followed by DAB substrate incubation (Supplementary Table 10) and counter-stained in hematoxylin before mounting in dibutyl-phthalate polystyrene xylene (DPX). Stained sections (n=6 per mouse) were imaged using an Olympus BX61 microscope. The positively stained area of dark brown color was quantified using Fiji (Image J). Each image was firstly color deconvoluted into H/DAB mode. Then the brown picture was selected, and the threshold was adjusted that matches the positive dark brown staining of the original image. The proportion of the total plaque area was indicated after drawing the region of interest from the internal elastic lamina to the vascular endothelium.”

Reviewer #2

The authors have leveraged their newly established and previously mouse model of unstable atherosclerotic plaque to conduct a proteomic analysis of unstable and stable plaques as well as normal vascular tissue. From their well powered mouse proteomic analysis, they identified multiple proteins that overlap with comparable conditions from human atherosclerotic plaques proteome analyses. Confirmation and biological validation were carried out on one particular target, S100A8/9, found to be significantly more prominent in the unstable plaque condition. Inhibition of this target promoted plaque stability in the carotid artery, confirming a possible mechanistic role of this protein and its associated inflammatory pathways in eventual plaque rupture. Prevention of plaque rupture could have wide reaching impact on major cardiac events, which remain the leading cause of death.

Overall, the experimental design is of high quality, the methods are very well articulated, and the interpretation / conclusions fit the data that have been presented.

Response: We would like to thank the Reviewer for their very positive and encouraging feedback. We addressed the Reviewer's comments on a point by point basis, which we believe strengthened our manuscript further.

The following are some minor comments:

1) Different arterial regions were used for each experimental group - and this may confound the comparisons in plaque state. What effect do baseline proteomic differences between aortic arch and carotid artery have on the comparison made between stable and unstable plaque (since stable plaques came from aortic arch while unstable are sampled from carotid artery)?

Response: We thank the Reviewer for pointing this out. It could be that the proteome of the aortic arch is different to the proteome of the carotid artery. The risk that we report on this

is mitigated by the following three points. 1) In our analysis we include a comparison with the plaque-free left carotid artery as control. 2) We include a comparison to human plaques. 3) In the comparison between carotid artery and aortic plaques, we do not see a major difference in the baseline proteomes, as the protein expression distribution and coefficient variations are very similar. We normalized our data at several levels to allow for comparison. Protein extracted from tissue was normalized, and peptides used for sample analysis were further normalized, in addition to single batch post-processing informatics normalization using MaxQuant (maxLFQ). We provide protein expression distributions for all sample groups, where we did not observe baseline proteomic differences in individual samples between the aortic arch, carotid arteries with unstable plaque, and carotid arteries without plaque (**Supplementary Figure 1A**). This is further supported by cross-group analysis of coefficient variation (**Supplementary Figure 1B**), which did not differ within our group, indicating that baseline proteomic variation (~20-22% coefficient variation) was very similar across groups.

In direct response to the Reviewer's comment we have now included the following text as a limitation in the discussion:

Page 12: "A potential limitation in our comparative proteomics approach is the fact that we cannot fully exclude that the difference in baseline proteome between the carotid artery (source of the unstable plaque in the TS model) and the aortic plaque (source of the stable plaque in the TS model) confounds the comparison of the plaque stability state. However, the following three points will mitigate this risk. 1) In our analysis we include a comparison with the plaque-free left carotid artery as a control. 2) We include a comparison to human plaques. 3) In the comparison between carotid artery and aortic plaques, we do not see a major difference in the baseline proteomes, as the protein expression distribution and coefficient variations are similar. We normalized our data at several levels to allow for comparison. Protein extracted from tissue was normalized, and peptides used for sample analysis were further normalized, in addition to single batch post-processing informatics normalization using MaxQuant (maxLFQ). We provide protein expression distributions for all sample groups, where we did not observe baseline proteomic differences in individual samples between the aortic arch, carotid arteries with unstable plaque, and carotid arteries without plaque (Supplementary Figure 1A). This is further supported by cross-group analysis

of coefficient variation (Supplementary Figure 1B), which did not differ within our group, indicating that baseline proteomic variation (~20-22% coefficient variation) was very similar across groups. In conclusion, the risk of the comparison of different localizations representing different phenotypes of plaques is a potential inherent confounding factor. However, we mitigated this risk as much as possible.”

2) Please elaborate on details for the mouse reference proteome used, what database was this proteome extracted from (Uniprot?) and are the entries reviewed (by the magnitude perhaps not?) and do they include canonical sequences only or do they also include isoforms of proteins?

Response: This information has now been updated in the Supplemental Methods file and provided below. Both SwissProt (reviewed) and TrEMBL (unreviewed) entries were employed.

Page 4 (Supplementary material) - “Peptide identification and quantification were performed using MaxQuant (v1.6.6.0) with its built-in search engine Andromeda⁶⁻⁸. Tandem mass spectra were searched as a single batch against the Mus musculus reference proteome (UniProt; UP000000589, 59,345 entries, Feb-2019; canonical protein sequence)...”

3) Supplementary methods - redundant entries for some of the methods.

Response: In response to the Reviewer’s comment, we have reviewed this and removed several areas of redundancy (sample perpetration, proteomic analysis, mass spectrometry) in the resubmitted manuscript.

4) Was normalization used in the maxLFQ algorithm, if so what method?

Response: Yes, maxLFQ (built in for label free quantitation) normalisation was performed for this study. The original study by Cox et al.,¹⁰. MaxLFQ is based on extracted ion current (XIC)-based approach and not spectral counting. This intensity-based approach takes into consideration the entire XIC for each sample for comparative purposes. The specifics of this algorithm are protected by the maxLFQ algorithms, however, the normalization approach has two main advantages (i) “delayed normalization”, which makes label-free quantification fully compatible with any up-front separation, and (ii) extracts the maximum ratio

information from peptide signals in arbitrary numbers of samples to achieve the highest possible accuracy of quantification.

We have now stated this in the Supplementary Material:

Page 5 (Supplementary material)- “The original MaxLFQ study by Cox et al.,¹⁰ is based on extracted ion current (XIC)-based approach and not spectral counting. This intensity-based approach takes into consideration the entire XIC for each sample for comparative purposes. The specifics of this algorithm are protected by the maxLFQ algorithms, however the normalisation approach has two main advantages of (i) “delayed normalization” which makes label-free quantification fully compatible with any up-front separation, and (ii) extracts the maximum ratio information from peptide signals in arbitrary numbers of samples to achieve the highest possible accuracy of quantification.”

5) also in supplementary methods - line 183-185, perseus does not quantify the proteins, it is for post-processing, dataset filtering, statistics and visualization. Protein quantification is completed prior to data upload into perseus. Please clarify in the text.

Response: We have made this change, replacing the word “quantify” with “analyze”

Page 5 (Supplementary material) - “Perseus (v1.6.14)¹¹ was used to analyze proteins whose expression was identified across multiple biological replicates (i.e., in at least 70% in at least one group). Statistical analyses were performed using Perseus, R programming, and GraphPad Prism, with unpaired two-sample Student’s t-test or one-way ANOVA performed (statistical significance defined at FDR<0.05)”

6) Figure text for some panels is very small and hard to read - furthermore resolution of figures appears low and they are fuzzy.

Response: We apologize for the low resolution and the small font size of the previously uploaded main and supplemental Figures. We have now updated the resolution/layout and adapted the font size of all figures.

REVIEWERS' COMMENTS:

Reviewer #1 (Remarks to the Author):

The authors have adequately addressed this reviewer's comments. I recommend that this article is accepted for publication with statistical review.

Reviewer #2 (Remarks to the Author):

The authors have sufficiently addressed my concerns with their clarifications throughout text and within the rebuttal.